# Genome-Wide Identification and Characterization of Long Non-Coding RNAs in Embryo Muscle of Chicken

**DOI:** 10.3390/ani12101274

**Published:** 2022-05-16

**Authors:** Lingbin Liu, Lingtong Ren, Anfang Liu, Jinxin Wang, Jianhua Wang, Qigui Wang

**Affiliations:** 1Key Laboratory of Biorheological Science and Technology, Chongqing University, Ministry of Education, Chongqing 400045, China; liulb515@swu.edu.cn; 2College of Animal Science and Technology, Southwest University, Beibei, Chongqing 400715, China; rltswu@163.com (L.R.); anfangliu@126.com (A.L.); 3Department of Neuroscience, University of Rochester Medical Center, Rochester, NY 14642, USA; jinxin_wang@urmc.rochester.edu; 4ChongQing Academy of Animal Sciences, Rongchang, Chongqing 402460, China

**Keywords:** chicken, embryo muscle development, lncRNAs, WGCNA, regulatory network construction

## Abstract

**Simple Summary:**

This study preliminarily determined the expression pattern of lncRNA during leg muscle development in the middle and late stages of embryo in chicken, and revealed regulatory lncRNAs and pathways associated with skeletal muscle development, providing a theoretical basis for further research on the molecular mechanism of muscle development in embryo.

**Abstract:**

Embryonic muscle development determines the state of muscle development and muscle morphological structure size. Recent studies have found that long non-coding RNAs (lncRNAs) could influence numerous cellular processes and regulated growth and development of flora and fauna. A total of 1056 differentially expressed lncRNAs were identified by comparing the different time points during embryonic muscle development, which included 874 new lncRNAs. Here, we found that there were different gene expression patterns on the 12th day of embryo development (E12). Herein, WGCNA and correlation analyses were used to predict lncRNA function on E12 through the screening and identification of lncRNAs related to muscle development in the embryo leg muscles of Chengkou mountain chickens at different times. GO and KEGG functional enrichment analysis was performed on target genes involved in cis-regulation and trans-regulation. An interaction network diagram was constructed based on the muscle development pathways, such as Wnt, FoxO, and PI3K-AKT signaling pathways, to determine the interaction between mRNAs and lncRNAs. This study preliminarily determined the lncRNA expression pattern of muscle development during the middle and late embryonic stages of Chengkou mountain chickens, and provided a basis to analyze the molecular mechanism of muscle development.

## 1. Introduction

Skeletal muscle is an essential tissue in vertebrates, accounting for 40% of body mass [1]. It is involved in various vital functions, including movement, coordination, and protection functions [2]. In addition, skeletal muscle of domestic animals is an important protein source for humans. Myogenesis is a complex biological process involving several gene regulation networks [2,3,4,5,6]. Muscle development is divided into embryonic muscle development and postnatal development [7]. Embryonic muscle development determines the state of muscle development and muscle morphological structure size. Differentiation and proliferation of muscle progenitor cells form myoblasts, which fuse to form multinucleated myotubes. The multinucleation causes the maturation of myotubes into myofibers with contractile properties [2,8]. The number of muscle fibers does not change after birth, and muscle weight increases in postnatal individuals, and hypertrophy of existing muscle fibers promotes the growth of skeletal muscle [9].

The muscle development process is determined through continuous analysis of its regulation mechanism. It involves multigene co-regulation processes, including MRFs (myogenic regulatory factors) [6], IGFs (insulin-like growth factors) [10], myocyte enhancer factor 2 (MEF2) [11], TGFβ (transforming growth factor β) [12], FoxO (forkhead box protein O) [13], Paxs (paired box gene family) [14], and SIXs (SIX gene family) [15]. These genes participate in different muscle development stages. Moreover, these genes form gene regulatory networks (through coordination or regulatory relationships) to regulate muscle development through different signaling pathways, including Wnt, BMP, mTOR, Notch, Hippo, and TGF-β pathways [6,16,17].

LncRNAs are non-coding RNAs longer than 200 nucleotides, with medium-low to moderate spatiotemporal expression patterns. They are categorized into five groups based on their position in the genome, including intronic, intergenic, sense, antisense, and bidirectional lncRNA [18]. They are involved in chromatin modification, transcription activation and inhibition, post-transcriptional mediation, and miRNA-induced molecular interference with gene expression, etc. Meanwhile, lncRNA also regulates different muscle development stages [19,20,21,22]. Several lncRNAs have been found due to the improved sequencing depth and breadth [22,23]. For instance, Gga-lnc-0181 is highly expressed in chicken pectoralis, providing a basis for using the chicken embryo as the model animal of embryonic development [24]. LncRNA regulates muscle development through a complex multigene regulation process based mainly on gene transcription, molecular sponge, and other regulatory mechanisms [25,26]. Most studies have focused on SYISL, Neat1, and Malat1 in recent years. SYISL, Neat1, and Malat1 are involved in muscle development via corresponding regulatory factors [27,28,29,30]. Myoblast proliferation and differentiation is crucial in the development of embryo muscle. Various lncRNAs are involved in myoblast development [8,31]. For instance, linc-2949 and linc1369 regulate the expression of c-Myc, thereby affecting the proliferation and differentiation fate of myoblasts [32].

Transcriptomics technology has been widely used in poultry breeding research due to the improved sequencing depth and decreased sequencing cost. For instance, 281 new intergenic lncRNAs have been found in White Leghorn chicken breast muscle [33]. A total of 539 differentially expressed lncRNAs have also been identified in Roman layer chicken (liver) before and after birth, revealing the potential molecular regulation mechanism of liver development in chicken, an indication that chicken livers provide energy to the body through the mitochondrial respiratory chain regulation pathways [34]. Although several lncRNAs have been identified via high-throughput sequencing, no study has reported on lncRNAs related to chicken muscle development. Therefore, the role of lncRNA in muscle development should be clarified.

As one of the local chicken breeds in China, the Chengkou mountain chicken has the distinctive characteristics of local high-quality chicken breeds, resistance to rough feeding, delicious meat, and high nutritional value [35,36]. However, it has similar problems as other local chicken breeds (slow growth rate and low meat yield). Investigating the complex regulatory mechanism of muscle development can help in the genetic improvement of the growth rate and yield of meat products. Moreover, understanding the complex regulatory mechanism of muscle development can help us to clarify the mechanism of atrophy and the regeneration process.

This study used Chengkou mountain chickens to explore the superiority of local chicken species in heritage performance by investigating four-stage chicken leg muscle (12-day embryo (E12), 16-day embryo (E16), 19-day embryo (E19), 21-day embryo (E21)). RNA sequencing was used to detect differentially expressed lncRNAs involved in embryo development. The study aimed to characterize the expression of differentially expressed lncRNA in muscle development and conduct the enrichment function and structural analysis of lncRNAs.

## 2. Materials and Methods

### 2.1. Chicken Embryo Incubation and Tissue Collection

This study used the Chengkou mountain chicken as the experimental animal. Chengkou mountain chicken breeding eggs were obtained from Chongqing Xuanpeng Agricultural Development Co. Ltd., Chongqing, China. The eggs were incubated at 37.8 °C and 55% humidity. Twelve chicken embryos were obtained at four time points (12, 16, 19, and 21 embryonic ages), with three replicates at each time point. The embryos were euthanized via cervical spine dislocations, and leg muscles were collected from the same sampling sites. The 12 samples were stored at −80 °C (wrapped in RNA protective solution (QIAGEN, Hilden, Germany)) for RNA extraction.

### 2.2. Construction of cDNA Library and Sequencing

Trizol reagent (TaKaRa, Dalian, China) was used to extract total RNA from chicken embryo leg muscle at the four stages, following the manufacturer’s protocol. Nanodrops were used to assess the quality and the purity of total RNA. rRNA was removed from the total RNA (Epicentre, Charlotte, NC, USA). The Illumina HiSeqTM 4000 was used for sequencing at the GENE DENOVO Biotechnology Co., LTD (Guangzhou, China). The FastQC (v0.11.4) was used to check the raw data quality. The original data were filtered as follows to ensure quality: (1) reads containing an adapter; (2) reads with an N-ratio greater than 10%; (3) all reads with A-bases; and (4) low-quality reads (the number of bases with mass value Q ≤ 20 accounting for more than 50% of the whole read), were excluded. The following analysis was based on high-quality sequencing. Clean reads were compared with the ribosome database of the species using the short reads alignment tool Bowtie 2. Reads with ribosomes were divided and compared to ensure no mismatches. Reserved unmapped reads were used for subsequent transcriptome analysis. HISAT 2 software was used for a comparative analysis based on the chicken genome (GCF_000002315.6). FPKM (reads per kilobase of exon model per million mapped reads) was used to determine sample expression. Sample repeatability was tested via principal component analysis.

### 2.3. LncRNA Transcript Analysis

LncRNA transcripts were filtered from embryo leg muscles at different stages, which were reconstructed using the stringtie software. In candidate transcripts, those with length ≥200 bp and number of exons ≥2 were filtered. The programs Coding Potential Calculator (CPC) and Coding-Non-Coding-Index (CNCI) were used to predict the coding ability of new transcripts. The reliable prediction of lncRNA transcripts was obtained from the intersection results. The new lncRNAs were classified based on their relative positions in the genome. The expression was normalized using FPKM (fragments per kilobase of exons per million mapped fragments) using the software RSEM. DESeq2 was used to reveal the significantly different lncRNAs, which, based on FDR < 0.05 and |log2FC| > 1. Infernal (v1.1.2), were used to classify all the predicted lncRNAs based on conservative sequences and secondary structures via multisequence alignment, secondary structure, and covariance model.

### 2.4. Interaction Analysis for mRNA and lncRNA Transcripts

RNAplex was used to determine the short interaction of two lncRNAs and predict the complementation fixation between antisense RNA and mRNA (mRNA data were obtained from [37]). The cis-regulation function of lncRNA was in relation to its neighboring protein-coding genes within 10 kb upstream or downstream of a gene. Further, trans-regulation was in relation to its co-expressed protein-coding gene attained through correlation analysis or co-expression analysis of lncRNA and protein-coding genes. Pearson correlation coefficient was used to estimate the correlation level. STRING (http://string-db.org (accessed on 10 February 2022)) was used to assess the differential gene protein interaction network. Cytoscape 3.4.0 was used to present interaction by building a network diagram.

### 2.5. Function Enrichment Analysis

The lncRNA–mRNA regulatory relationship was constructed to analyze the function of target genes further to clarify the mechanism of lncRNA involved in muscle development. We identified genes of each term in the mapping database GO (http://www.geneontology.org/ (accessed on 10 February 2022)) and calculated the number of genes in each term and created a GO function gene list and statistics. A hypergeometric test identified GO entries significantly enriched in genes compared with the entire genome background. KEGG Pathway was used to identify pathways significantly enriched in genes compared with the entire genome background via a hypergeometric test. The most important biochemical metabolic pathways and signal transduction pathways were determined through significant enrichment pathway analysis. The corrected *p*-value of ≤0.01 was considered statistically significant.

### 2.6. Verification and Statistical Analysis

Herein, six lncRNAs were used to verify the sequencing results via RT-qPCR. Primers were designed by Primer Premier (Appendix A). RNA reverse transcription and real-time fluorescence quantitative PCR were performed by referring to [37]. ACTB was used as a housekeeping gene for qPCR. The relative lncRNA expression was calculated via the 2^−^^△△CT^ method [38], and data are expressed as mean ± standard deviation of the mean. Statistical significance of the RT-qPCR quantitative expression data was tested by performing two-tailed unpaired *t*-tests (SPSS Inc., Chicago, IL, USA). Graphics were plotted using GraphPad Prism 7 (GraphPad Software, San Diego, CA, USA). *p*-value was used to evaluate the data significance level.

## 3. Results

### 3.1. lncRNA Sequencing

Herein, 12 cDNA libraries (E12-1, E12-2, E12-3, E16-1, E16-2, E16-3, E19-1, E19-2, E19-3, E21-1, E21-2, E21-3) from embryo leg muscle were constructed to obtain the comprehensive and accurate lncRNA transcripts. A total of 8402 lncRNA transcripts (encoded by 5562 genes) were identified in all sequencing libraries. The predictive ability of transcript coding detected 5768 known lncRNAs and 874 new lncRNAs (Figure 1A). The lncRNAs were divided into five categories based on their location in the genome: 5156 intergenic long-chain non-coding lncRNAs (intergenic lncRNAs), 935 bidirectional long-chain non-coding lncRNAs (bidirectional lncRNAs), 352 intron long-chain non-coding RNA (intronic lncRNAs), 1734 antisense long non-coding RNAs (antisense lncRNAs) and 150 sense long non-coding RNAs (sense-overlapping lncRNAs) (Figure 1B). FPKM showed that lncRNAs had different expressions (Figure 1C and Appendix A). LncRNAs with transcript lengths of less than 200 bp were negligible (Appendix A). The distribution of transcripts on chromosomes also exhibited a similar trend (Figure 1D). PCA was used to analyze and calculate the Pearson correlation coefficient between samples based on the known expression of lncRNA in each sample to understand the repeatability between samples and exclude outlier samples. This ensured the reliability of sample selection and the accuracy of the analysis. LncRNA transcripts were divided into four different components. The Pearson correlation coefficient was used to determine the relationship among the lncRNAs, then presented as a heat map (Appendix A).

### 3.2. Differentially Expressed lncRNAs during Embryo Muscle Development

This study analyzed the differentially expressed lncRNAs (DE lncRNAs) at different stages (E12, E16, E19, and E21) (Fold change ≥ 2, FDR < 0.05) to find the key lncRNAs involved in the muscle development of the leg muscles of Chengkou mountain chickens. FPKM was used to measure the transcript expressions. A total of 1056 DE lncRNAs were identified by comparing the different time points during embryo muscle development. The E12vsE16, E12vsE19, E12vsE21, E16vsE19, E16vsE21 and E19vsE21 comparison groups had 321, 565, 683, 125, 322 and 112 DE lncRNAs, respectively (Figure 2A). The down-regulated transcripts were highest during the four stages. The expression quantity clustering analysis was used to determine the expression patterns of DE lncRNAs at different time points. The cluster analysis results show that the intra-group repetitions at each time point were clustered, indicating a smaller difference within the study group than between the groups, reflecting the reliability of the results. There were two types of clustering between E12 and the other three time periods (E16, E19, E21) (Figure 2B), indicating that day 12 had different expression patterns from the other time points. This further suggested a significant transition between E12 and E16. Venn diagrams were developed using differential comparison groups to further explore the key lncRNAs involved in embryo muscle development (Figure 2C). However, no lncRNA was involved at the four time points.

### 3.3. Weighted Gene Co-Expression Analysis (WGCNA) of LncRNAs

WGCNA was used to find the key modules associated with the different stages of embryonic muscle development to further detect lncRNAs involved in the regulatory role. Eight soft thresholds were used to ensure that the module conformed to scale-free distribution (Appendix A). LncRNAs were divided into 20 modules based on their expression and named using different colors (Figure 3A). Different degrees of correlation were found between modules (Figure 3B). We counted the number of lncRNAs appearing in 20 modules, and the statistical results are shown in Figure 3C. Correlation analysis was used to cluster the lncRNAs in the module and construct the expression connections among lncRNAs. The darker the color of each point (white→yellow→red), the stronger the connectivity between the two genes corresponding to the row and column, and the stronger the Pearson correlation (Appendix A). Association analysis was conducted using the characteristic values of modules and different traits to determine the most relevant modules at different time points for further analysis. Modules and traits had significant correlation differences. For instance, E12, E16, E19 and E21 were most significantly correlated with turquoise modules (r = 0.88, *p* < 0.01), gray modules (r = 0.81, *p* = 0.001), magenta modules (r = 0.61, *p* = 0.03), and green-yellow modules (r = 0.59, *p* = 0.04), respectively (Figure 3D).

### 3.4. Prediction of lncRNA Potential Regulation

LncRNA affects mRNA expression through various action modes. This study predicted the potential regulation relationship between DE lncRNA and mRNA (including the cis-regulation and trans-regulation) based on previous mRNA studies to explore how lncRNAs interact with their target genes to regulate chicken muscle development and identify key molecules in this process. E12 was selected for subsequent analysis of lncRNAs enriched in turquoise modules since it had a developmental difference from the other time points. We found a total of 1249 lncRNAs in the turquoise module, and obtained 469 lncRNAs that play a regulatory role in E12 after intersecting with differential lncRNAs (Figure 3E). Further, we explored the role of 469 lncRNAs in embryonic muscle development. First, we detected that 213 lncRNAs influenced 278 mRNAs via cis-regulation by analyzing the relationship between lncRNA and neighboring protein-coding genes. A total of 304 lncRNA cis-regulatory interaction relationships were predicted. The interaction between lncRNAs and co-expressed genes was analyzed using trans-regulatory. A total of 86,178 trans-regulatory interaction relationships were detected between 392 lncRNAs and 3493 mRNAs.

### 3.5. Function Analysis of lncRNAs and Co-Expressed Genes

Functional enrichment analysis of the target genes regulated by lncRNA was conducted to explore the possible functions of lncRNA under the focus module. Based on the turquoise module’s significant correlation with E12, we found 469 differentially expressed lncRNAs that might play an important role in this stage, and further predicted the mRNA associated with them. Functional analysis of the target genes involved in cis-regulation and trans-regulation was also conducted. The GO enrichment analysis of the cis-regulation of lncRNAs targets showed several muscle-development-related terms (Appendix A), including the muscle cell migration, the muscle structure development, the muscle cell differentiation, and the regulation of muscle tissue development, which were enriched in E12. GO enrichment analysis of trans-regulation of lncRNAs detected several muscle-development-related GO terms in E12 (Figure 4A), such as skeletal muscle cell differentiation, skeletal muscle tissue regeneration, skeletal muscle cell differentiation, regulation of fast-twitch skeletal muscle fiber contraction, skeletal muscle cell proliferation, and skeletal muscle adaptation.

A Kyoto Encyclopedia of Genes and Genomes (KEGG) pathway analysis for cis-regulation and trans-regulation targets was conducted to further assess how lncRNAs influence mRNAs. The Yersinia infection pathway was identified as the most significantly enriched pathway in cis-targets of lncRNAs. The PI3K-Akt signaling pathway, mTOR signaling pathway, and the ECM-receptor interaction that affects development progress were also enriched in cis-targets of lncRNAs (Appendix A). Furthermore, the cell cycle was the most significantly enriched pathway in the trans-regulation lncRNAs. The Wnt signaling pathway, a key signaling pathway during development, was also significantly enriched. Moreover, the KEGG pathway analysis showed that PI3K-Akt, MAPK, Hippo, AMPK, TGF-beta, and FoxO signaling pathways were enriched in the trans-targets of lncRNAs (Figure 4B). A regulatory network of the pathways involved in the trans-regulation targets was then constructed (Figure 5). The network diagram showed that the cell cycle, as a key pathway of ontogenesis, occurred throughout the whole life process. Therefore, the expression of the key regulatory genes of this pathway and the lncRNAs that regulate their expression can significantly impact the whole development process.

### 3.6. Co-Expression Network Establishment

A few pathways involved in embryo muscle development were identified based on the differential expression analysis, WGCNA analysis, and function analysis results. The regulatory networks for different pathways involved in development were separately constructed. Correlation analysis was used to construct the network diagram of target gene interaction. Target gene prediction found that the Wnt signaling pathway was involved in the development at E12. The main genes included DKK2, SRFP1, SRFP2, SRFP4, and CTNNBIP1. The regulatory network between these genes and lncRNAs was constructed based on the association analysis results (Figure 6A). The regulation network of FoxO, AMPK, and PI3K-Akt pathway-related genes, such as *PLK4*, *FBXO32*, *TBC1D1*, and *MYB*, were also constructed (Figure 6B,D,E). MYOG is essential in muscle development. However, its mechanism in muscle development is unclear. Herein, several lncRNAs had a regulatory effect on MYOG at E12 (Figure 6C), indicating that these lncRNAs are involved in muscle development.

### 3.7. Validation of Candidate lncRNAs

This study screened several lncRNAs, including XR_212291.3, XR_001471472.2, XR_001466502.2, MSTRG.17928.1, XR_001466777.2, and XR _003077657.1, to reveal the key lncRNAs associated with embryonic muscle development (primer information given in Appendix A). RT-qPCR verification for the selected DE lncRNAs generated findings consistent with the lncRNA-seq findings (Figure 7). Notably, the results are consistent with the sequencing results, confirming the reliability of the sequencing data.

## 4. Discussion

Although heredity, nutrition, and environment regulate muscle growth and development, genetics has received much attention. Myogenesis involves muscle formation through the coordination of transcriptional cascade reactions and signaling pathways that guide cell proliferation, differentiation, migration, and morphological changes. Animal muscle development is mainly divided into embryonic and postnatal stages. The embryonic stage is the key stage of muscle fiber formation, where muscle fiber morphology and structure are formed. Muscle fiber development is composed of primary and secondary muscle fiber. Previous studies have shown that primary muscle fiber formation occurs in the first 3–5 days of the embryonic stage in chickens, while secondary muscle fiber development occurs from the 15th day of the embryonic stage until after birth [38]. The quality of embryonic muscle development is related to the whole process of muscle development.

Muscle development is a multigene regulation process, including protein-coding and non-coding genes. Although previous studies have elucidated the myogenesis process, the specific mechanism of muscle development is unclear. The extensive application of omics technology in muscle development research has gradually enriched the gene pool related to myogenesis. Research has shown that lncRNA is involved in muscle development. lncRNAs have different action modes in the nucleus and cytoplasm due to their different locations. Most nuclear lncRNAs regulate the chromatin state by guiding chromatin modification elements to act on specific gene sites. Meanwhile, self-regulation of nuclear lncRNA transcripts also negatively impacts gene expression. Cytoplasmic lncRNAs have sequence complementarity with transcripts from the same locus or independent sites. They mainly control gene expression by regulating mRNA stability, translation, and post-translational modifications [39].

The Chengkou mountain chicken is an excellent local breed in Chongqing, China, that can meet the demand of modern people for high-quality livestock and poultry products. However, this breed has not been widely used due to genetic limitations. Herein, RNA sequencing was used to analyze the embryonic skeletal muscle of the Chengkou mountain chicken at the full transcriptomic level, screen the lncRNAs related to muscle development, and construct a gene regulatory network for embryonic muscle development. A total of 8057 lncRNAs were detected in four time points. Differential gene expression analysis filtered 1056 lncRNAs, which were potentially involved in the embryo muscle development regulation.

The development stages of poultry embryos are relatively short and have substantial time differences. Different time points have different development forms, and the identification of specific expressions of lncRNA and mRNA at each stage is essential in the analysis of the development regulation in poultry. Herein, WGCNA analysis was used to explore the specific expression at the four stages. Every stage had a significance module, and candidate lncRNAs were found at the target module. E12, E16, E19, and E21 were significantly correlated with turquoise, gray, magenta, and green-yellow, respectively. E12 was used for subsequent analysis.

To date, multiple studies have shown that lncRNA is involved in muscle development. Dum (developmental pluripotency-associated 2 (*Dppa2*) upstream binding muscle lncRNA) silences its neighboring gene *Dppa2* by recruiting Dnmt1, Dnmt3a, and Dnmt3b to promote myoblast differentiation [40]. Six1 is involved in sensory systems and skeletal muscle. Moreover, lncRNA located in 432 bp upstream of the Six1 (lncRNA-Six1) can encode micropeptides to activate Six1 [41]. A new lncRNA (434 nt long) located downstream of Cdkn1b, originating from 4 kb 3′ to the *Cdkn1b* gene, is involved in cis-regulation [42]. However, trans-regulation analysis has shown that lncRNA function is based on gene co-expression. SYISL regulates PRC2 expression, thus changing the myogenesis process [29]. Malat1 regulates myogenic differentiation and muscle regeneration via MyoD modulation [28]. This study predicted that 392 lncRNAs and 3493 mRNAs were involved in trans-regulation. MyoD is a key transcription factor involved in muscle development. MyoD knockout disrupts the normal muscle development process and is mainly reflected in the abnormal cell cycle. MyoD activates the expression of the neighboring gene LncMyoD. LncMyoD then directly binds to IMP2 (IGF2-mrN-binding protein 2), negatively regulating the imp2-mediated translation of n-Ras, C-MyC, and other proliferative genes [43]. H19 is associated with lncRNA LOI (loss of imprinting), preventing the normal down-regulation of p53 activity, inhibiting the expression of the IGF1 receptor, and impacting the activation of the AKT/mTOR signaling pathway, eventually affecting muscle differentiation and hypertrophy [44].

The potential mechanisms of lncRNA and mRNA were predicted to determine the functions of candidate genes. There were three types of interaction between lncRNA and mRNA, including antisense, cis-regulation, and trans-regulation. LncRNAs can regulate the expression of neighboring mRNAs (10 kb upstream and downstream) (cis-regulation). Herein, 213 lncRNAs were predicted to influence 278 mRNAs’ expression. GO and KEGG analysis found some development-related entries and pathways and screened the key lncRNAs and their target genes. Herein, multiple biological-process-related items mainly involved in transcriptional regulation and biosynthesis during development, including cells, cell cycle, skeletal muscle cell differentiation, skeletal muscle tissue regeneration, skeletal muscle cell differentiation, regulation of fast-twitch skeletal muscle fiber contraction, skeletal muscle cell proliferation, and skeletal muscle adaptation, were significantly enriched in GO. Additionally, several biological metabolic pathways related to muscle development, including Wnt, ECM-receptor interaction, FoxO, AMPK, MAPK, and PI3K-Akt signaling pathways, were enriched in KEGG. In the study on the dynamic changes in lncRNA expression in the breast muscle of the Gushi chicken at different periods, the authors reflected similar results to this study [45].

Wnt protein is a secreted growth factor belonging to the family of conservative glycoproteins rich in cysteine. Many biological processes rely on ligand-mediated signaling pathways, from organ formation during embryonic development to adult stem cell homeostasis [46]. In addition, the Wnt signaling pathway is highly conserved and is essential in early embryonic development, organogenesis, and tissue regeneration. It also plays a crucial role in muscle development. Previous studies have shown that Plectin binds to DVL-2 and forms a protein complex to activate the typical Wnt signaling pathway. This inhibits the apoptosis of C2C12 myoblasts, ultimately promoting myoblast differentiation and proliferation [47]. KLF5 may regulate chicken skeletal muscle atrophy through the Wnt/β-catenin signaling pathway [48]. This study conducted target gene prediction of DE lncRNAs and found key genes in the Wnt pathway, such as DKK2, SRFP1, etc., involved in muscle development. Moreover, their related lncRNAs may regulate muscle development.

The PI3K-Akt signaling pathway is involved in muscle development, mainly at the stage of myoblasts. Herein, many lncRNA target genes were enriched in the PI3K-Akt signaling pathway. Previous studies have shown that fibroblasts can protect myoblasts from endogenous apoptosis-related differentiation via the activation of the β1 integrin/PI3K/Akt pathway [49]. Insulin-like growth factor (IGF-1) is the most common upstream signaling molecule in the PI3K/Akt pathway. IGF-1 activates the PI3K/Akt pathway mainly by binding to the IGF-1 receptor. It is essential in myoblast proliferation and differentiation. Besides inducing myoblast proliferation in vitro, IGF-1 can also increase chicken embryo skeletal muscle satellite cells [50]. The PI3K-AKT pathway can also regulate cell proliferation by regulating the expression of downstream target proteins mTOR/p70S6K/ULK. These results suggest that gene interaction is a complex gene regulatory network, which affects muscle development [51]. P38-mitogen-activated protein kinase (MAPK) is one of the several signaling pathways that drive skeletal muscle metabolic adaptation. The MAPK family plays a key role in complex cellular processes, such as proliferation, differentiation, and development, by regulating the cell cycle and other proliferation-related proteins. Three members of the MAPK family have been identified: extracellular signal-regulated kinase (ERK), JNK/SAPK, and P38 MAPK [52]. Li [45] revealed for the first time the dynamic changes in lncRNA expression in Gushi chicken breast muscle at different periods. The MAPK signaling pathway is one of the most common enrichment pathways in different control groups, revealing that the MAPK signaling pathway plays a crucial role in muscle development. Studies have shown that GGA-Mir-3525 regulates SMSCs’ proliferation and differentiation by targeting the chicken P38/MAPK signaling pathway [53]. Several studies have shown that PI3K/AKT and MAPK/ERK1/2 signaling pathways act together to regulate myoblast proliferation and muscle development. Moreover, the MAPK/ERK1/2 signaling pathway is crucial in protein synthesis during the proliferation of skeletal muscle cells. Cycle mechanical stretch (15% and 20%) can regulate the proliferation of rat L6 myoblasts and is related to the regulation of PI3K/Akt and MAPK (P38 and ERK1/2) expression and IGF-1/IGF-1R activity [54].

## 5. Conclusions

In conclusion, this study constructed the lncRNA sequencing library of the Chengkou mountain chicken, identifying 8057 lncRNAs and 1056 DE lncRNAs via differential expression analysis. WGCNA and association analysis were used to determine the target genes of related lncRNAs. GO and KEGG functional analysis of target genes showed that WNT and PI3K-Akt pathways were related to muscle development. This study provides a theoretical basis for further research on the molecular mechanism of skeletal muscle development in embryo.

## Figures and Tables

**Figure 1 animals-12-01274-f001:**
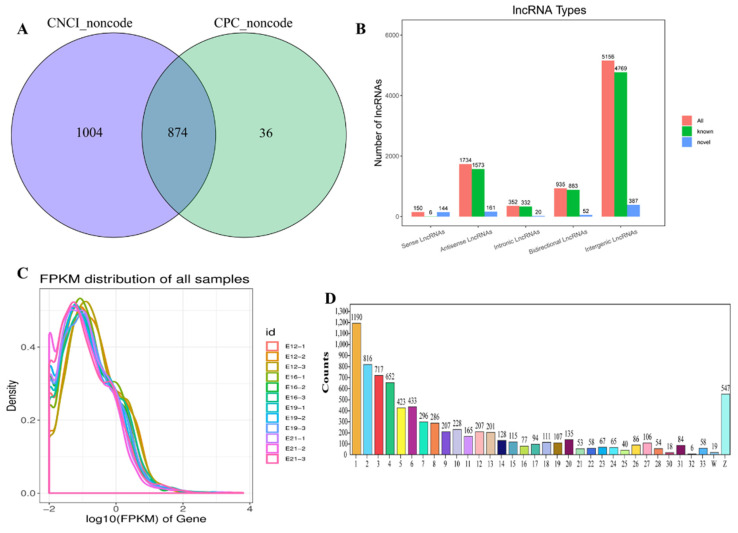
Overview of RNA sequencing in the muscle stages. (**A**) Venn diagram of CPC and CNCI (the prediction of novel lncRNAs). (**B**) The types of lncRNAs. Abscissa represents lncRNA types, and the ordinate represents lncRNA quantity, respectively; (**C**) The distribution of FPKM for lncRNAs; (**D**) the transcripts’ number distribution of chromosomes for lncRNAs. Abscissa represents chromosome type, and the ordinate represents lncRNA quantity, respectively.

**Figure 2 animals-12-01274-f002:**
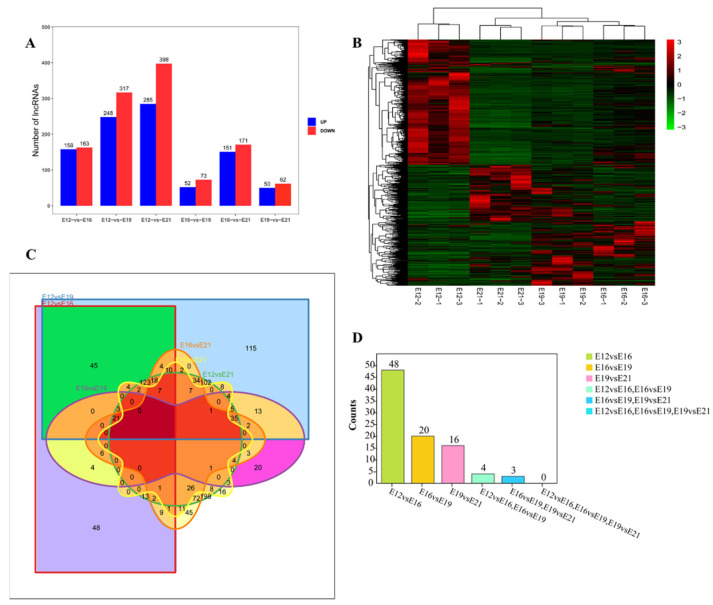
Differential expression of lncRNA analysis. (**A**) The number of lncRNAs differentially expressed at different times, including up-regulation and down-regulation; (**B**) differential lncRNA cluster analysis; (**C**) Venn diagram of six comparisons of differentially expressed lncRNAs; (**D**) stage difference gene statistics.

**Figure 3 animals-12-01274-f003:**
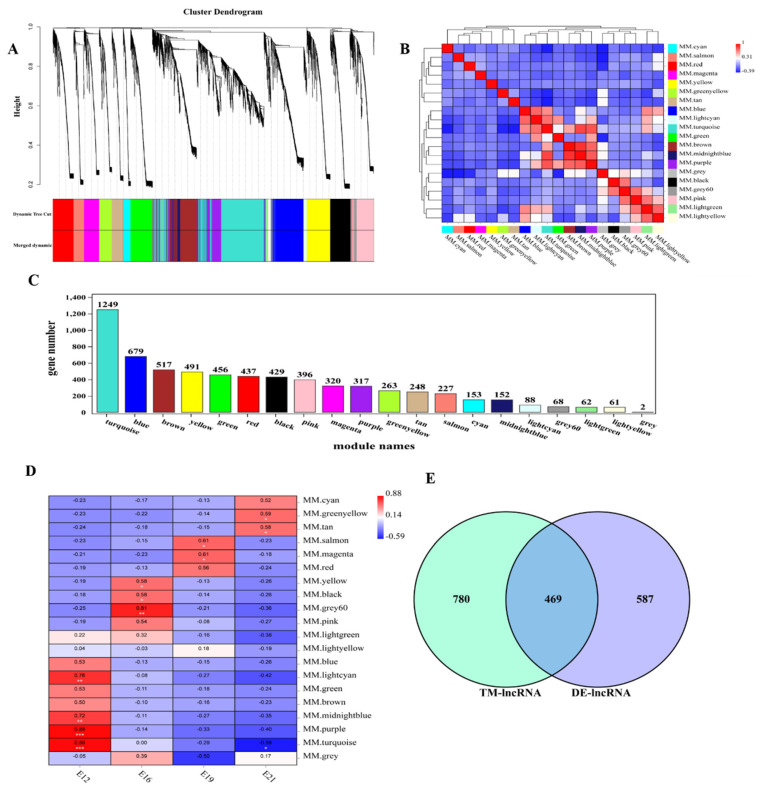
Weighted gene co−expression network analysis of lncRNAs. (**A**) Module eigenvalue clustering; (**B**) correlation analysis of modules; (**C**) lncRNA statistics of each module. Different colors of the axes represent different modules, respectively; (**D**) trait correlation analysis; (**E**) the differential expression gene screening from E12 module.

**Figure 4 animals-12-01274-f004:**
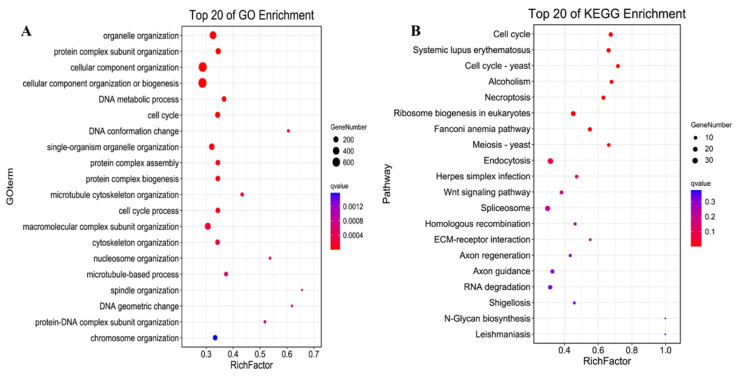
The functional enrichment analysis of the trans-regulation lncRNAs. (**A**) the top 20 significantly changed GO terms of mRNAs; (**B**) top 20 significantly changed pathways associated with mRNAs.

**Figure 5 animals-12-01274-f005:**
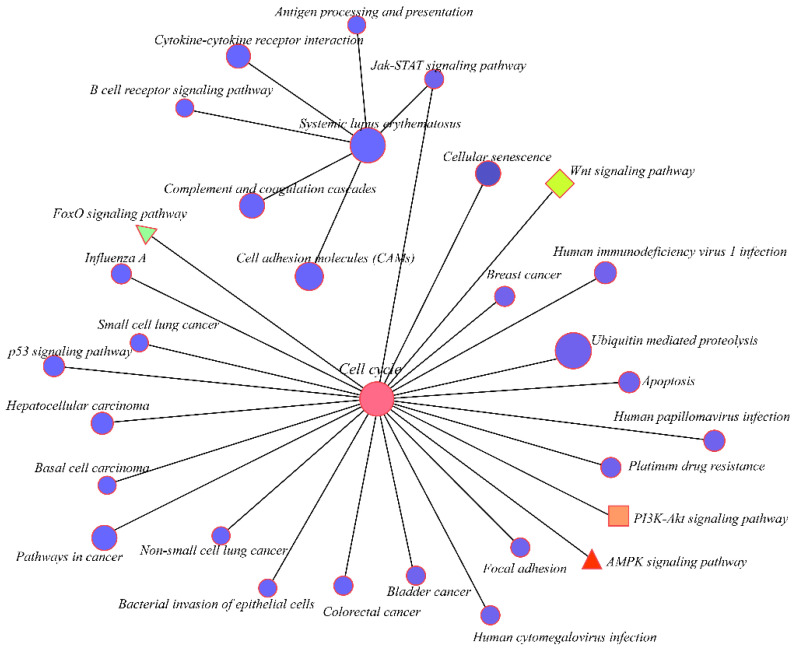
Correlation of the top 20 KEGG pathways. Different colors represent the different enriched pathways.

**Figure 6 animals-12-01274-f006:**
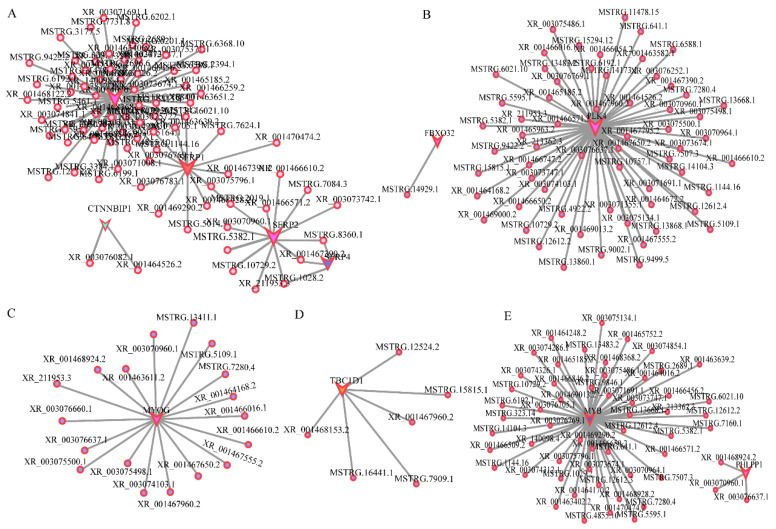
Interaction of lncRNA and mRNA. (**A**). The Wnt signal pathway; (**B**). The FoxO signal pathway; (**C**). MYOG; (**D**). The AMPK signal pathway; (**E**). The PI3K-AKT signal pathway. Polygon represents mRNA, and the circle represents lncRNA, respectively. Different colors represent different mRNAs, and polygon size represents the degree of enrichment, respectively.

**Figure 7 animals-12-01274-f007:**
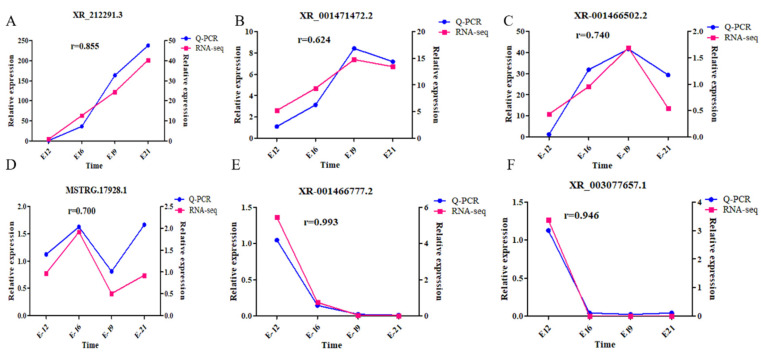
Target lncRNA validation. (**A**) XR_212291.3; (**B**) XR_001471472.2; (**C**) XR_001466502.2; (**D**) MSTRG.17928.1; (**E**) XR_001466777.2; and (**F**) XR_003077657.1.

## Data Availability

The raw data have been submitted to the National Center for Biotechnology Information (NCBI) Sequence Read Archive (SRA), and the accession number is PRJNA674456 [37].

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
