# Peer review of "Genome-Wide Identification and Characterization of Long Non-Coding RNAs in Embryo Muscle of Chicken"

_animals, 2022, doi:10.3390/ani12101274_

Round 1
Reviewer 1 Report
The manuscript is well written, Introduction section is clear and guides the reader through the key points of the study. Although conducted on a breed with main diffusion in China, the study reveals new information about lncRNA in muscle tissue in chicken. So, the results are quite interesting for scientific step forward in this field. Below, the authors may find specific comments that IMO will improve the quality of the manuscript before publication.
- Please add lines throughout all the manuscript to facilitate review.
Abstract
- The journal guidelines refer to a maximum of 200 words. Please shorten the abstract (is now more then 300 words).
Methods
- Which was the RIN obtained from extracted RNA?
- "Nucleic acid tester" is a software or an instrument? Why not QUBIT or nanodrop for checking quality? Please specify.
- Please add information related to sequencing: PE or SE, length of the fragments, depth of sequencing...
- Add information related to alignment: uniquely mapped reads rate (%),...
- Annotation: which software has been used? Which release? Please add.
Discussion:
Authors limited their Discussion to Chengkou Mountain Chicken, that is surely correct, but maybe adding parallels with other species in which lncRNA in muscle have been observed may add a more wide idea of the utility of this study.
In the same direction, is this study useful also for chicken as a species? Which are the differences between breeds that can be highlighted in term of muscle lncRNA (if any)? Please add some considerations on that.
Figures
Please improve the quality of the figures. For instance, in figures 1B ,2B and 3B the text on the axis and legend need to be increased; figures 1D and 1C appear overstretched. In supplementary figure with correlation plot the text is too small and also it appears blurred.
Author Response
1.According to the reviewer's comments, the summary was compressed and revised(after line 19).
2.In view of the problems in the materials and methods proposed by reviewers, we made corresponding modifications according to the problems in the comments, which improved the scientificity and accessibility of the article(after line 108).
3. The problems in the discussion have also been revised according to the reviewers' comments(after line 333).
4.Modified the problems existing in the figure of manuscript and supplementary figure(after line 561) .

Reviewer 2 Report
The manuscript provides a detailed bioinformatic analysis of the lncRNA of the muscles of chicken during embryogenesis that might be helpful for the genetic selection of these chickens.
I have some comments on the validation of the six lncRNAs that were performed in this study.
For the primers used in the qPCR, the authors are advised to revise the correct accession numbers as they have been updated.
For the primers used in the qPCR (Table S1, methods, and the results text), the authors are advised to revise the correct accession numbers as they have been updated.
I checked the primers sets and found these problems:
XR_005855851.1 instead of XR_001471472.2
XR_005853884.1 instead of XR_003077657.1
XR_212291.4 instead of XR_212291.3, also check the correct F primer to be “CTCTCCTGCTGAGCAAGCCATCG”
XR_005860270.1 instead of XR_001466777.2, also check the correct F primer to be “TGGCATAATGGGTGAACAGGGAAG”
For the last two sequences “XR_001466502.2 & MSTRG.17928.1”, I cannot find any matching of the provided sequences in BLAST analysis.
Why did you choose beat actin as a housekeeping gene for analysis of LncRNA? You should choose the best candidate.
Author Response
1.Regarding the primers raised by reviewers, we carefully reviewed the occurrence of lncrnas and primers in the article. Regarding the inconsistency between the primers you proposed and lncRNA, we found that the problem did exist, but this problem was mainly because the gene sequences we sequenced were used when we designed the primers, instead of the existing sequences in NCBI, and there were certain differences between the sequenced sequences and the original ones.The primers obtained by sequencing sequence were verified by RT-QPCR, and the results showed a high correlation with RNA-seq data, which further confirmed the accuracy of our sequencing data.(In response to the questions raised by reviewers, we uploaded the sequencing data of relevant lncRNAs).
2. As for the selection of internal reference genes, a large number of previous literatures have reported that the selection of internal reference genes for lncRNA is consistent with that for mRNA, and our study mainly focuses on the expression of lncRNA during embryonic muscle development, so we chose ACTB as the internal reference gene for verification.

Reviewer 3 Report
This is a well rounded paper looking at differences of lncRNAs expression in different stages of muscle development. In total, 8,402 lncRNA transcripts were identified, which encoded by 5,562 genes. The authors have also validated some of the identified lncRNAs using qpcr. Apart from that, weighted gene co-expression network analysis (WGCNA) showed that the turquoise module was significantly correlated with E12, indicating that the target genes of lncRNAs could be in the turquoise module. Overall there are no concerns.
Author Response
Thank you very much for your recognition of this paper.
Reviewer 4 Report
Dear authors,
This study tries to study the role of lncRNAs in the chicken embryo muscle development, analyzing the lncRNA differentially expressed in different stage of development of Chengkou Mountain chickens. The manuscript is well written and structured; however, the authors should have included line numbers to facilitate review. Another hand, the introduction provides sufficient background, and the cited references are relevant to the research. Some major changes are necessary before publication:
- “The number of muscle fibers does not change after birth. Muscle weight increases in postnatal individuals and hypertrophy of existing muscle fibers promote the growth of skeletal muscle”- Add references that support these sentences.
- “They are categorized into five groups based on their position in the genome”-Specify which groups.
- “Full transcription analysis can reveal the difference in gene expression between atrophic ovaries and normal ovaries, detect lncRNAs, miRNA, and genes related to ovarian atresia, and construct a regulatory network related to ovarian development”- This information does not present relevant data about the area of interest of this manuscript. I recommend to authors, to eliminate this sentence.
- “…thus providing a theoretical basis for the pathogenesis and treatment of various muscle disease”- The authors refer to muscle diseases in humans? Specify. If so, perhaps the authors should have used an animal model with human-like embryonic development.
- “The relative lncRNA expression was calculated via the 2-ΔΔCt method…”- The authors should add the reference of this method (Livak, 2001).
- “p<0.05 and p<0.01 were considered statistically significant and extremely significant, respectively”- The p-value is fixed before performing the statistical analysis and, in statistics, there is no concept of "extremely significant". A comparison is or is not significant, depending on the p-value parameter chosen a priori. Therefore, this sentence does not make sense. Authors should indicate the p-value set for statistical analysis and stick to it in their calculations and reported results.
- Other minor changes are necessary too.
Author Response
According to the comments of reviewers, we modified the content of the article one by one.
1. We added relevant references. (line 47, 155)
2. We improved the content of the article. (line 58, 86, 158)

Round 2
Reviewer 2 Report
The manuscript is now acceptable.
Please follow the authors' guidelines and the provided manuscript template.
Reviewer 4 Report
No comments